# Silicon-Based On-Chip Tunable High-Q-Factor and Low-Power Fano Resonators with Graphene Nanoheaters

**DOI:** 10.3390/nano13101636

**Published:** 2023-05-13

**Authors:** Qilin Hong, Jinbao Jiang, Siyu Zhou, Gongyu Xia, Ping Xu, Mengjian Zhu, Wei Xu, Jianfa Zhang, Zhihong Zhu

**Affiliations:** 1College of Advanced Interdisciplinary Studies, National University of Defense Technology, Changsha 410073, China; qlhong_95@163.com (Q.H.);; 2Hunan Provincial Key Laboratory of Novel Nano-Optoelectronic Information Materials and Devices, National University of Defense Technology, Changsha 410073, China; 3Institute for Quantum Information and State Key Laboratory of High Performance Computing, College of Computer Science and Technology, National University of Defense Technology, Changsha 410073, China

**Keywords:** graphene transparent electrodes, Fano resonators, high Q-factor, silicon photonics

## Abstract

Tunable and low-power microcavities are essential for large-scale photonic integrated circuits. Thermal tuning, a convenient and stable tuning method, has been widely adopted in optical neural networks and quantum information processing. Recently, graphene thermal tuning has been demonstrated to be a power-efficient technique, as it does not require thick spacers to prevent light absorption. In this paper, a silicon-based on-chip Fano resonator with graphene nanoheaters is proposed and fabricated. This novel Fano structure is achieved by introducing a scattering block, and it can be easily fabricated in large quantities. Experimental results demonstrate that the resonator has the characteristics of a high quality factor (∼31,000) and low state-switching power (∼1 mW). The temporal responses of the microcavity exhibit qualified modulation speed with 9.8 μs rise time and 16.6 μs fall time. The thermal imaging and Raman spectroscopy of graphene at different biases were also measured to intuitively show that the tuning is derived from the joule heating effect of graphene. This work provides an alternative for future large-scale tunable and low-power-consumption optical networks, and has potential applications in optical filters and switches.

## 1. Introduction

In the past decade, silicon photonic integrated circuits (PICs) have been regarded as one of the most promising platforms for realizing high-performance and multifunctional on-chip photonic devices. Microring resonators (MRRs), as one of the most fundamental components in silicon PICs, are widely used in the area of modulation [1,2], filters [3] and sensing [4,5]. As silicon PICs are scaled up, high-performance MRRs usually require a high Q-factor, low energy consumption, and a large tunable range.

Thermo-optic tuning is an effective way to realize tunability owing to the large thermo-optic coefficient of silicon [6] (∼1.8$\;\times \;10^{-4}$/K at 1550 nm). The drawbacks of thermal tuning are millisecond responses and temperature sensitivity, which hinders high-speed modulation. However, in certain application fields, such as optical neural networks [7,8] and quantum information processing [9] where the scale of the network and the number of photonic devices are more important than modulation speed, the thermal tuning of microcavities may be an appropriate low-cost, easily fabricated, and reliable modulation method. Conventional thermal tuning is generally achieved by sputtering TiN metals [10] on a thick silica upper-cladding layer (∼1 μm) that could isolate metals from the silicon core, so the evanescent field is not be affected by the great optical losses of metals. However, the small thermal conductivity of silicon oxide (1.4 W/(m·K)) and the required 1 μm thick spacer determine the low heating efficiency of metallic microheaters. Even worse, conventional thermal tuning is prone to thermal crosstalk due to the nondirectional thermal transport in the silica upper-cladding layer and becomes pronounced in high-integration networks [11,12]. For instance, in the optical 4 × 4 silicon router [11], heating one microring simultaneously gives a phase shift to the nearby ring, making them lose their independence. A conventional strategy to enhance heat efficiency is to incorporate free-standing silicon resonators with an undercut structure, which could reduce thermal diffusion around the core waveguide [13]. Nevertheless, this approach has the disadvantages of the weak mechanical strength of the free-standing structure, a complex fabrication process, and poor reliability in high-density networks. Another interesting strategy is using the part of silicon that contains no electromagnetic field to transfer heat [14,15] since the thermal conductivity of silicon (150 W/(m·K)) is much higher. The issue is that such a structure needs intricate manufacturing techniques to generate heater regions with varying doping levels; there must also be an affect to not affect the electromagnetic field confined in the microrings as much as possible.

Graphene, a semi-metallic two-dimensional (2D) material composed of a single layer of carbon atoms, has been extensively investigated due to its diverse and charming optoelectronic properties [16,17,18,19,20], including broadband absorption of less than 2.3% per layer from the visible to the far-infrared range [18,19] and carrier mobility of up to 200,000 cm^2^/V·s [21,22]. In the field of thermology, graphene also shows high intrinsic thermal conductivity on account of the long-wavelength phonon transportation in 2D crystal lattices [23,24], proved by an experimental value of up to 5300 W/(m·K) in room temperature [25]. On the basis of these unique properties, many studies regarded graphene as having “transparent” electrodes and showed its advantages [26,27,28,29,30]. Yu et al. [30] placed graphene in direct contact with silicon to thermally tune the microdisc resonator at around 1550 nm, and show the improvement in heating efficiency (∼4 mW for a resonance shift). However, since microcavities with a high quality (Q) factor are very sensitive, the optical losses of graphene (although much smaller than those of metals) become hard to ignore when graphene directly comes into contact with the microcavities, leading to a low spectral resolution (∼1 nm line width) and variation in the Q-factor at different power levels. In addition, the inevitably increased temperature of the central part of microdiscs causes unnecessary power consumption.

To further improve the performance of this type of devices, in this paper, we present a thermally tunable resonator that combines graphene nanoheaters with high heating efficiency and Fano resonators with low switching-energy requirements. The Fano resonator is achieved via the incorporation of microring resonators (MRRs) with a scattering rectangle block that could be easily fabricated using deep ultraviolet (DUV) lithography. Experimental results demonstrate that the Fano resonator with a graphene nanoheater requires only ∼1 mW for “on/off” state switching, and has a narrow line width of ∼0.05 nm, outperforming previous on-chip thermally tunable silicon devices [30]. The results also demonstrate that the Q-factor of the resonator remains stable with a change in voltage, which is essential for practical applications. The oscilloscope measurements of the temporal responses show that 90% of the rise and fall times of this tunable Fano resonator are 9.8 and 16.6 μs, respectively. The thermal imaging and Raman spectroscopy of graphene at different biases were also measured to intuitively show that the tuning is derived from the joule heating effect of graphene. This low energy consumption and high-Q structure may be a promising building block for on-chip integrated photonics, and could be applied to optical filtering [31,32] and switching [33,34].

## 2. Design and Mechanism

Fano resonance is a widely known phenomenon that occurs when a narrow discrete constructive or destructive state interferes with the continuum band of states [35]. Compared to normal Lorentzian resonance, asymmetric Fano resonance shows a sharper slope and narrower wavelength range from minimal (reflectivity) to maximal (reflectivity) transmittance [36,37]. These characteristics allow for Fano resonance to behave better in a variety of optical applications, such as sensing [38], switching [39] and nonlinearity [40]. Specific to on-chip microring, although there are many structures realizing Fano resonance, including Mach–Zehnder interferometers [41], Fabry–Perot (FP) cavities [42], and they are either complex or at the expense of compact footprints. Introducing an air-hole in the bus waveguide is compact, but it is difficult to fabricate because the diameter of the air hole, for example, should be larger than 360 nm [37], which leaves the margin of a standard 500 nm wide waveguide at less than 70 nm. Furthermore, the proximity effect becomes more pronounced at small exposure nodes, hindering the air-hole structure from acquiring the desired shape, even with EBL [40].

To achieve asymmetric Fano line shapes, as shown in Figure 1a, we designed a microring structure with a scattering block that is compact and could be easily fabricated with DUV lithography (90/180 nm for the majority of the MPW foundry). Using the theory of temporal coupled mode [43,44], the analytical expression of the transmittance can be derived as follows (see Appendix A):(1)$$T={\left|t_{B}-\frac{2\left(r_{B}\sqrt{\gamma_{1}\gamma_{2}}e^{i\left(\theta_{2}-\theta_{1}\right)}+t_{B}\gamma_{2}\right)}{i\left(\omega_{0}-\omega \right)+\gamma_{1}+\gamma_{2}+\gamma_{A}}\right|}^{2}$$
where ω is the incident frequency, $\omega_{0}$ is the resonant frequency of the cavity, $\gamma_{1/2}$ are the decay rates towards the bus waveguide, $\gamma_{A}$ is the intrinsic cavity loss rate, and $t_{B}$ and $r_{B}$ are the corresponding direct transmission and reflection coefficients of the amplitudes, respectively. $\kappa_{1/2}$ are the complex coupling coefficients between the ports and the cavity that satisfy $\kappa_{j}=\sqrt{2\gamma_{j}}e^{i\theta_{j}}\;\left(j=1,2\right)$.

To plot the theoretical transmission spectra, we set the parameters as follows: *R* = 30 μm, $n_{\mathrm{eff}}=2.45$, $t_{B}=0.8e^{i\pi /6}$, $\gamma_{1}=24.5\times 10^{10}$ Hz and $\gamma_{A}=1\times 10^{10}$ Hz. Since the structure was mirror symmetry, it was reasonable to assume that $\kappa_{1}=\kappa_{2}$. For the Lorentzian line shape (MRRs without scattering block), there should be no reflection for the bus waveguide, so $t_{B}=1$ and other parameters were kept unchanged for the comparison. As depicted in Figure 1b, the asymmetric line shape was realized through the designed structure, and its frequency shift (orange, 1.34 nm) from the maximal transmittance (0.986, dashed line) to the minimal transmittance was significantly smaller than the Lorentzian one (blue, 4.98 nm), even though they had the same Q-factor. If the slope rate (SR) is defined as $(T_{\max}-T_{\min})/\Delta \lambda$, then the SRs of the Fano and Lorentzian line shapes are 0.736 and 0.198, respectively. This nearly fourfold improvement means that achieving thermo-optic modulation based on the proposed structure could further reduce power consumption. In order to verify the correctness of the theory, we also built a three-dimensional model through COMSOL Multiphysics software, and calculated the transmittance in different incident frequencies (see Appendix A).

## 3. Methods

**Sample fabrication**. The Si photonic platform was prepared by the MPW foundry (IMECAS, China) with standard processes of DUV lithography and inductively coupled plasma etching, followed by the deposition of 2 μm thick silica as the upper-cladding layer. To avoid the crack of graphene caused by the height difference of the waveguides, we then used a CMP machine (LOGI TRIBO) to planarize the top surface of the silica and thin it down to about 180 nm. The thickness of the top silica layer was measured with the ellipsometer. To remove silica nanoparticles introduced by the CMP step, the sample was cleaned using ultrasonic cleaners for 15 min. Graphene and hBN flakes were mechanically exfoliated from highly oriented graphite and hBN crystals, respectively. The hBN and graphene were stacked with the van der Waals assembly technique [45,46] (see Appendix A) and then transferred directly to the top of the ring through an optical microscope (50×). Before the transfer process, the sample was exposed to oxygen plasma (200 W, 360 s) to ensure good adhesion between the 2D materials and the waveguide. The hBN–graphene stack was patterned via an electron-beam lithography (EBL, TESCAN MAGNA) system using PMMA (950k A6) resistance, and etched with ICP in an O_2_:CHF_3_:Ar (10:20:10) SCCM environment for 2 min to expose the graphene edge.Metal electrodes were fabricated via electron-beam evaporation and the lift-off of 5 nm Cr/50 nm Au.

**Experimental setup and measurements**. Single-mode fiber arrays were utilized to couple in and out light via 70 nm deep shallow-etched gratings. The input port was connected to a tunable continuous-wave laser (YENISTA T100S-HP) with a fiber polarization controller to maximize the input power. The output power of the tunable laser was generally fixed at 1 mW, and the coupling efficiency of the gratings was approximately 8 dB. For the measurement of static characteristics, the output fiber was connected to a high-sensitivity power meter (THORLABS PM100). A DC signal was applied to two probes with a sourcemeter (KEITHLEY 2636B). For thermo-optic temporal response measurement, the output fiber was first connected to a photodetector (THORLABS PDA50B2) and then a real-time digital oscilloscope (YOKOGAWA DL6154). The RF signal was generated by an arbitrary waveform generator. The signal was split in the oscilloscope to monitor the source (see Appendix A).

## 4. Results and Discussion

The presented high-Q-factor Fano resonator with a graphene nanoheater consisted of an especially designed silicon-on-insulator (SOI) platform with an hBN–graphene heterostructure, as illustrated in Figure 2a. Silicon waveguides with a cross-section of 500 × 220 nm (w × h) were embedded into the silica to operate in single transverse-electric (TE) mode at a wavelength of ∼1550 nm (inset I, Figure 2a). To achieve this structure, the SOI platform was first deposited with 2 μm thick silica, planarized with chemical mechanical polishing (CMP) techniques, and thinned down to ∼180 nm (t). The radius of the fabricated microring was R=30μm, and the gap of the side-coupled region is 200 nm. The size of the scattering block was 250 × 1000 nm, as shown magnified in inset II, Figure 2a. Both sides of the bus waveguide were connected with grating couplers to couple the light in and out. For the electrical part, bilayer graphene (checked via Raman spectroscopy, Appendix A) and hBN stacks were realized following pick-up fabrication techniques [47], and connected with metal electrodes to apply the bias (inset III, Figure 2a, see Section 3 for more details). Optical microscopy and scanning electron microscopy (SEM) images of the fabricated device are shown in Figure 2b, revealing that the Fano resonator was good (Inset I, Figure 2b) and the hBN-graphene stacks were precisely transferred onto the top of the ring (inset III, Figure 2b).

Figure 2c depicts the measured transmission spectra of the Fano resonator before and after the transfer of the hBN–graphene stack. We utilized a tunable laser as the external light source, and a power meter to acquire normalized transmittance (see Section 3 for more details). Before the transfer, the line width Δλ of the transmission spectrum (defined as $\left|\lambda_{\mathrm{peak}}-\lambda_{\mathrm{dip}}\right|$) was ∼0.025 nm, and the modulation depth was ∼90%. The free spectral range (FSR) of the ring (not shown here) and the corresponding Q-factor were ∼3 nm and ∼61,800, respectively (estimated as λ/Δλ). This transmission spectrum showed quite a high Q factor and a very sharp slope in the context of the SOI platform, demonstrating the correctness of the previous theory. The dip approaching zero indicated that the proposed system was close to the critical coupling. With the transfer of the hBN–graphene stack, because of the graphene absorption, the line width increased to ∼0.05 nm and the Q-factor reduced to ∼31,000. However, this Q-factor was still high enough [30]. Correspondingly, the modulation depth decreased to ∼39% because the additional losses put the system in the region of undercoupling.

To verify the device’s thermal tunability, a DC bias was applied to the graphene using a sourcemeter through the pads. Since the resistance of graphene is higher than that of other parts of the electric circuit, the joule heating effect was mainly concentrated on the graphene. With the increment in voltage, the heat generated by graphene diffused into the core of the waveguide through the top silica layer. Consequently, the effective refractive index ($n_{\mathrm{eff}}$) of the ring increased due to the positive thermo-optic effects of silicon. According to the resonant equation of microring $2\pi n_{\mathrm{eff}}R=m\lambda$, where *m* is the resonant order, the resonant wavelengths redshifted. As plotted in Figure 3a, the resonant wavelength redshifted from 1547.94 to 1548.11 nm when the bias increased from 0 to 4 V. Correspondingly, the whole power consumption rose from 0 to 3.56 mW, calculated with U·I (Figure 3b). Moreover, the Q-factor and transmittance values stayed almost the same during this process, which is important for practical applications.

The response time of thermo-optical tuning systems is another crucial characteristic parameter influenced by thermal coupling and isolation. To measure the modulation speed of this Fano resonator, a square signal was applied to the graphene with an arbitrary waveform generator (AWG), and the incident wavelength was fixed at 1548 nm (see Section 3 for more details). The high and low levels of the square signal were 2.4 and 1 V, respectively. As depicted in Figure 3c, the output power of the Fano resonator had the maximal value at the voltage of 1 V and was reduced to the minimum when the bias was 2.4 V. The frequency of the square signal was then raised to 1 kHz. The corresponding optical signal is plotted in Figure 3d. The blue line is the directly applied bias from the AWG to the oscilloscope, and the orange one is the optical signal detected by the photodetector. The 90% rise time and the decaying time of the thermal tuning were about 9.8 and 16.6 μs, respectively.

To show the tuning derived from the joule heating effect of graphene, Figure 4a illustrates the thermal image of graphene at various voltages recorded with an infrared thermal imaging microscope (OPTOTHERM IS640). As the graphene voltage rose from 0 to 10 V, the dashed area heated up rapidly and led to a slight temperature increase in the SiO_2_ surroundings and electrode sections. However, constrained by the spatial resolution of the microscope (5 μm/pixel), the graphene part could not be photographed more clearly. Although we compensated for the emissivity of different materials via temperature correction before capturing, for low-emissivity materials, software emissivity compensation alone is susceptible to significant errors due to variation in surface emissivity and errors in the control temperature measurement process. To overcome this problem, the average temperature of the dashed area was calibrated with the Raman spectroscopy of graphene (see Appendix A). As shown in Figure 4b, we gave the variation of graphene temperature by successively increasing and decreasing the bias voltage over a period of time. The overall graphene temperature was stable without significant fluctuations, and the repeatability between voltage and temperature was good.

There are two methods to further improve the performance of a Fano resonator. First, the optical losses of monolayer graphene can be electrically or chemically suppressed by improving the Fermi level of graphene to Pauli blocking regions [48]. Chemical predoping such as adding tungsten oxyselenide [49] or immersing in an AuCl_3_ solution [50] could be adopted before the transfer of graphene. Second, etching graphene into a bowtie shape could increase the current density of the narrowest part [51]. This helps in concentrating most of the heat to directly above the microring to avoid wasting power in situations where the contact resistance of graphene–metal is large.

## 5. Conclusions

In conclusion, we proposed and demonstrated low-power-consumption thermally tunable on-chip Fano resonators with graphene nanoheaters. The Fano resonators comprised an MRR and an especially designed scattering block, and are compact and easy to prepare. The graphene was laid on top of the planarized microring to act as a heating electrode. The fabricated Fano resonator showed energy consumption characteristics of down to ∼1 mW, and a spectral resolution as narrow as ∼0.05 nm. The low power consumption was mainly due to two aspects: the designed asymmetric line shape greatly reduced the required energy of state switching, and using graphene as the nanoheaters reduces the thickness of spacers that are required in conventional metal electrodes. Our results provide a promising building block for massive on-chip integrated devices, and may have applications in optical filters and switches.

## Figures and Tables

**Figure 1 nanomaterials-13-01636-f001:**
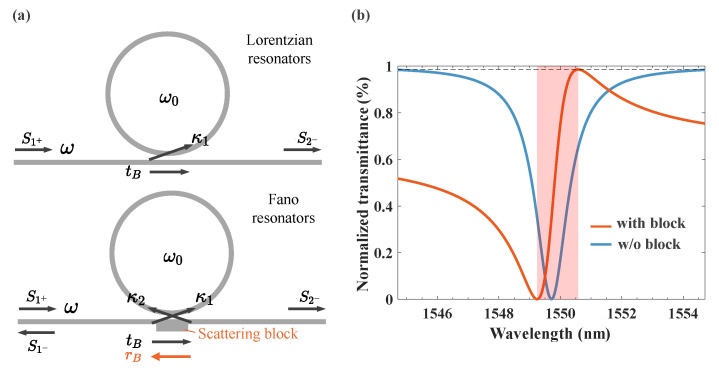
(**a**) Schematic illustration of the coupling of a waveguide to a resonator and the identification of the important parameters of the system. The scattering block represents the designed silicon rectangle block to realize Fano resonance. (**b**) Normalized theoretical transmission spectra of MRRs (orange line) with and (blue line) without a protruding scattering block.

**Figure 2 nanomaterials-13-01636-f002:**
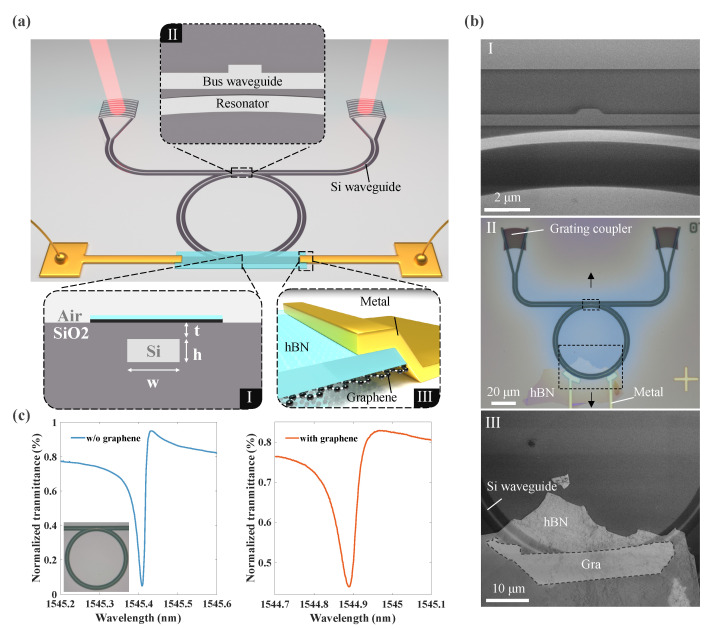
(**a**) Schematic illustration of the Fano resonator with graphene nanoheaters. I: a cross section of the structure. II: expanded view of the resonator with a scattering block. III: side view of an hBN–graphene stack connected with metal. (**b**) SEM and microscope images of the Fano resonator. I: SEM image of the upper dashed box in inset II. II: microscopy image of the fabricated Fano resonator. Bilayer graphene in this substrate cannot be seen directly. III: SEM image of the lower dashed box in inset II before metal evaporation. (**c**) Transmittance of the Fano resonator before and after the transfer of graphene. The inset is the microscopy image without graphene.

**Figure 3 nanomaterials-13-01636-f003:**
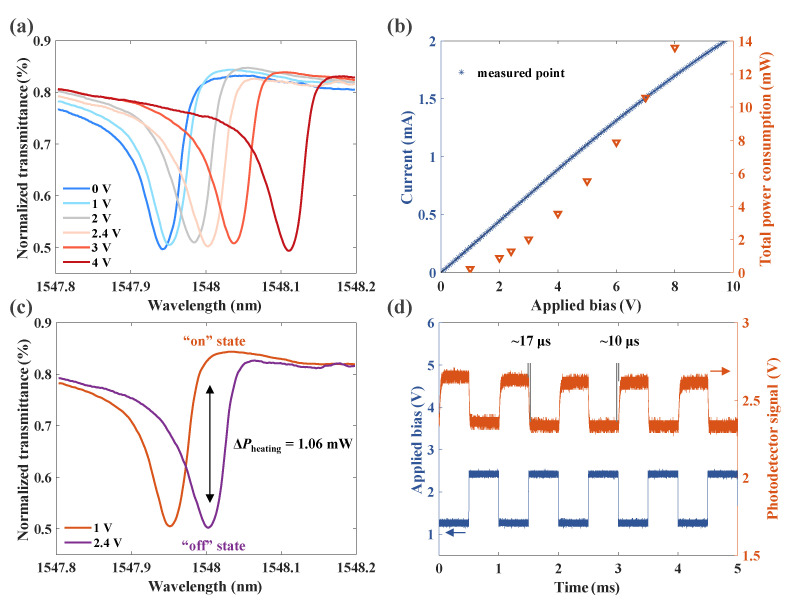
Characterization of the fabricated high-Q Fano structure with a transparent graphene electrode. (**a**) Measured transmission spectral responses of the proposed thermally tunable Fano resonator with DC bias from 0 to 4 V. (**b**) Measured I–V curve and the total power consumption of the whole circuit. Asterisks represent the measured points. (**c**) Minimal power consumption of the proposed structure for an “on/off” state switch. The incident wavelength was fixed at $\lambda_{0}=1548$ nm, and the bias was changed between 1 and 2.4 V. The corresponding switching power was 1.06 mW. (**d**) Temporal responses of the thermo-optical tuning system. The blue line represents square voltages applied to the graphene nanoheaters with a frequency of 1 kHz. The orange line represents the optical response signal detected by a photodetector.

**Figure 4 nanomaterials-13-01636-f004:**
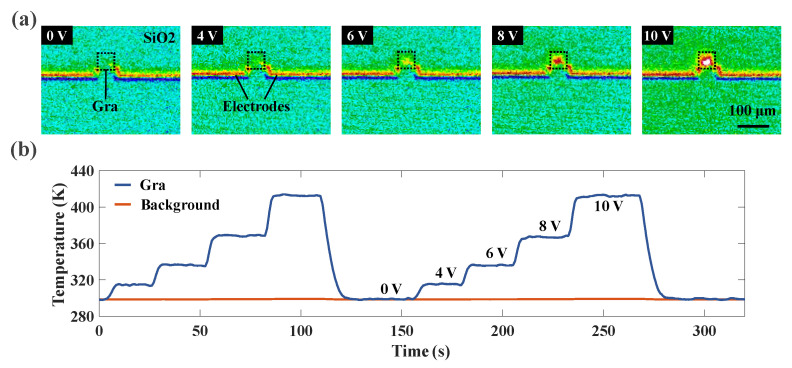
(**a**) Infrared thermal image of graphene with varying applied bias. (**b**) Graphene lattice temperature with changing voltages over a period of time.

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
