# Peer review of "Silicon-Based On-Chip Tunable High-Q-Factor and Low-Power Fano Resonators with Graphene Nanoheaters"

_nanomaterials, 2023, doi:10.3390/nano13101636_

Round 1

Reviewer 1 Report

The authors report a study of a fano resonance resonator fabricated in SOI technology. This device is then thermally tuned to show the resonance properties. The main innovation is the use of Graphene on BN as the heater element. The devices are characterised in the usual way, by a combination of Raman scattering, IR thermal imaging and transmission measurements with and without heating. The article is well written and presented with no obvious shortcomings. The article is of interest to working both in the fields of integrated photonics and 2D materials. Its not clear that the use of 2D materials represents a clear improvment over eg TiN heaters. But the possibility of lower energy is topical and warrents publication of the article.

Author Response

Thank you for your recognition of our work and writing.

Reviewer 2 Report

In this work, Authors reported about fabricated silicon-based on-chip Fano resonator with graphene nanoheaters. The presented results demonstrate that the resonator has the characteristics of high quality-factor and low state switching power. Thermal imaging and Raman spectroscopy of graphene at different bias demonstrate that the tuning can be derived from the Joule heating effect of graphene. Authors suppose, this work provides an alternative for large-scale tunable and low-power-consumption optical networks in the future, and has potential applications in optical filters and switches. Manuscript is appropriate for this journal and I would recommend this work for publication.

Author Response

Thanks for your comment. We are appreciated for your support of our work.

Reviewer 3 Report

The submitted work studies thermally tuneable resonator that combines graphene nano-heaters with high heating efficiency and Fano resonators with low switching-energy requirements. I have the following comments thereabout:

1. Temperature tuneability of the micro-cavity properties has both advantages and drawbacks. Among the drawbacks one may list slow tuning, sensitivity of the micro-cavity properties upon the ambient temperature and so forth. An analysis is necessary of advantages and weak points of thermal tuning of micro-cavity properties.

2. It is claimed that optical filters and switches may be created on the basis of the reported results. The Authors should provide possible examples of such implementations, at least in general.

3. It is necessary to describe in more detail the radiation used in the experiments. Was it CW or pulsed radiation (pulse duration, etc.), what was the radiation source, and so forth.

If the above-listed observations are taken into consideration in a further revision of the manuscript, it may be published in Nanomaterials.

Round 2

Reviewer 3 Report

In response to my observations, important information was added to the manuscript that made it more interesting and comprehensible. My comments have been fully addressed by the Authors in the revised manuscript, which may be now published.